# Protective Effects of the Food Supplement Flexovital in a Model of Cardiovascular-Kidney-Metabolic Syndrome in Mice

**DOI:** 10.3390/nu16234105

**Published:** 2024-11-28

**Authors:** Lucas Rannier Ribeiro Antonino Carvalho, Maria Tydén, Miho Shimari, Zhengbing Zhuge, Tomas A. Schiffer, Matheus Morais de Oliveira Monteiro, Jon O. Lundberg, Eddie Weitzberg, Daniel C. Andersson, Bengt Fellström, Mattias Carlström

**Affiliations:** 1Department of Physiology and Pharmacology, Karolinska Institutet, 17165 Solna, Sweden; lucas.carvalho@ki.se (L.R.R.A.C.); miho.shimari@ki.se (M.S.); zhengbing.zhuge@ki.se (Z.Z.); tomas.schiffer@ki.se (T.A.S.); monteirommo@gmail.com (M.M.d.O.M.); jon.lundberg@ki.se (J.O.L.); eddie.weitzberg@ki.se (E.W.); daniel.c.andersson@ki.se (D.C.A.); 2Department of Medical Sciences, Nephrology, Uppsala University, 75236 Uppsala, Sweden; maria.tyden@medsci.uu.se (M.T.); bengt.fellstrom@medsci.uu.se (B.F.); 3Cardiology Unit, Theme for Heart, Vascular and Neuro, Karolinska University Hospital, 17164 Stockholm, Sweden

**Keywords:** food supplementation, *Rhodiola rosea*, beetroot, arginine, citrulline, cardiovascular-kidney-metabolic syndrome

## Abstract

**Background/Objectives:** The prevalence of cardiovascular-kidney-metabolic (CKM) syndrome is increasing rapidly, and cardiovascular complications pose significant risks in individuals with kidney disease and metabolic dysfunction. Understanding the mechanisms of CKM disorders is crucial, as is the discovery of novel preventive treatments. This study aimed to examine the therapeutic effects of a specially formulated nitric oxide-enhancing food additive in a mouse model of CKM syndrome induced by unilateral nephrectomy (UNX) in combination with chronic Western diet (WD) feeding. **Methods**: C57BL/6J mice underwent UNX and were fed a WD high in salt, sugar, and fat for 12 weeks, compared to sham-operated mice on standard chow. One group of UNX+WD mice received Flexovital (FLX), a food additive containing extracts of *Rhodiola rosea* and beetroot, and the amino acids L-arginine and L-citrulline. CKM parameters were assessed both in vivo and ex vivo alongside histological and biochemical analyses. **Results**: The UNX+WD mice showed an increase in body fat mass, the fat/lean mass ratio, and adipocyte area, of which most were significantly reduced by FLX. Elevated fasting glucose levels were also reduced by FLX, which tended towards improving glucose clearance. Elevated arterial blood pressure and endothelial dysfunction in UNX+WD mice were significantly reduced by FLX. FLX improved GFR and reduced glomerular and tubular injuries in UNX+WD mice. Additionally, FLX increased the P/O ratios of oxidative phosphorylation in the isolated renal mitochondria of UNX+WD mice. **Conclusions**: In this model of CKM syndrome, FLX effectively prevented the onset and progression of CKM dysfunctions induced by UNX+WD, as well as the associated organ injuries. These promising results highlight the need for validation in upcoming human trials.

## 1. Introduction

Chronic kidney disease (CKD) is characterized by persistent kidney dysfunction, often indicated by albuminuria and a reduced glomerular filtration rate (GFR), categorized into disease stages 1 to 5 [1]. It is an escalating health problem, estimated to affect >10% of the global population, and one of the leading causes of mortality worldwide [2]. Cardiovascular disease (CVD) is highly prevalent in CKD patients, with risk escalating with the severity of CKD [3,4]. The risk elevation of coronary artery disease is also present in patients with earlier stages of CKD [5]. Moreover, CKD is associated with incident heart failure (HF), both with a preserved and reduced ejection fraction (EF), and linked to increased mortality [6]. It is acknowledged that patients with advanced CKD have an excess mortality that can be attributed to CVD, with the highest mortality rates among patients on dialysis [7,8]. In a recent presidential advisory, the American Heart Association (AHA) introduced the definition of cardiovascular-kidney-metabolic (CKM) syndrome, emphasizing the interplay between metabolic risk factors, chronic kidney disease (CKD), and the cardiovascular system, which profoundly impacts both morbidity and mortality [9].

The prevalence of obesity is increasing globally, as well as its implications for augmenting the risk of CKD and CVD [10]. Several population-based studies show a correlation between obesity and the development and progression of CKD as well as albuminuria, even in individuals without kidney disease [11,12,13]. Glomerular hyperfiltration as a response to increased metabolic demands due to high body weight is thought to be one of the mechanisms contributing to CKD in obese individuals [11].

Obesity contributes to an increased cardiovascular risk [14,15], and is directly linked to risk factors like hypertension, type 2 diabetes, sleep disorders, and dyslipidemia [16,17,18,19]. Visceral adiposity leads to an inflammatory state which promotes atherosclerosis [20]. In CKD patients, abdominal obesity is associated with higher mortality, even after adjusting for hypertension and diabetes in those with end-stage kidney disease (ESKD) and kidney transplants [21,22]. A meta-analysis suggested that eliminating obesity could prevent up to 30% of CKD cases across the entire industrialized world [23].

The traditional Western diet (WD), with processed foods high in animal protein, fat, sugar, and sodium, with low fruit and vegetable intake [24], is linked not only to increased fat accumulation but also to an elevated risk of microalbuminuria and accelerated kidney function decline [25]. Population-based studies show high sodium intake is associated with increased blood pressure [26,27]. Experimental animal studies show that high salt intake worsens renal injuries after unilateral nephrectomy [28,29], while low salt intake is associated with reduced kidney damage [30]. In the U.S., high salt and animal protein consumption is believed to contribute to kidney disease progression [24], while reducing animal protein and increasing fruits, vegetables, and fiber may delay ESKD [31].

Excess cardiovascular risk is a major contributor to the increased mortality and morbidity observed in both T2D and CKD populations. Further research is crucial to better understand the underlying pathophysiological mechanisms driving CKM syndrome and to optimize nutritional interventions aimed at preventing the onset and progression of CKD and CVD. Reduced nitric oxide production and signaling, commonly associated with oxidative stress, are widely acknowledged as important contributors to CVD and its associated renal and metabolic comorbidities [32,33]. The aim of this study was to investigate the effects of a specially formulated nitric oxide-enhancing food additive (Flexovital^®^) in a newly developed mouse model of CKM syndrome, induced by a reduced nephron number and chronic feeding with an unhealthy Western diet (WD) [34]. Flexovital, administered as capsules (up to a maximum of 4 per day in humans), contains *Rhodiola rosea* extract (100 mg), beetroot extract (100 mg), and the amino acids L-arginine (175 mg) and L-citrulline (125 mg), along with small amounts of magnesium (25 mg) and vitamin C (25 mg) per capsule.

## 2. Material and Methods

### 2.1. Animals and Experimental Design

This study was approved (Protocol code: 17816-2021; Approval date: 13 November 2021) by the Regional Institutional Animal Care and Use Committee at Karolinska Institutet in Stockholm, Sweden and performed according to the NIH guidelines and with the EU Directive 2010/63/EU for the conduct of experiments in animals. Young male C57BL/6J mice (Janvier Labs, Le Genest-Saint-Isle, France) were obtained and housed in the animal facility at Comparative Medicine, Karolinska Institutet. The mice were kept under controlled conditions for temperature and humidity, and in a 12-h light/dark cycle, with food and water freely available.

The mice arrived at the animal department (Comparative Medicine, Karolinska Institutet) at the age of 3 weeks, were acclimatized for one week and then subjected to either a sham or unilateral nephrectomy (UNX) surgery at 28 days of age, as recently described [34]. Upon complete recovery from surgery, after 7 days, the mice were randomized to three distinct experimental groups: (1) a sham operation with a standard rodent chow (RD, CRM(P) 801722, SAFE, Rosenberg, Germany) (sham n = 14); (2) a disease model group combining the UNX with the Western diet (WD) enriched in fat, sugar, and salt (4% NaCl) custom prepared by Research Diets Inc. (New Brunswick, NJ, USA) (UNX+WD, n = 13); and (3) a disease model group (UNX+WD) treated with a special food additive (Flexovital^®^—BioconceptAB—Stockholm, Sweden) in the drinking water (UNX+WD+FLX n = 8). The total number of experimental animals used in this study was n = 35. Flexovital was given in a dose of 25 g/L and contains extracts of *Rhodiola rosea* (12.8%) and beetroot (12.8%), the amino acids L-arginine (22.4%) and L-citrulline (16.0%), and minor additives of magnesium citrate (3.2%) and vitamin C (3.2%). All groups were followed for 12 weeks, and the gain in body weight, food and water intake, and other biological parameters of cardiovascular-kidney-metabolic function were monitored. A schematic illustration of the design of the experimental study is shown in Figure 1.

### 2.2. Metabolic Function

*Body Composition Analysis*: Fat and lean masses were measured by dual-energy X-ray absorptiometry (DEXA) using a Medikors InAlyzer densitometer (MEDIKORS Inc., Seongnam, Republic of Korea). Body composition was assessed using the total fat and lean mass in grams, as well as their respective percentages relative to overall body mass.

*Glucose and Insulin*: The metabolic variables analyzed were blood glucose levels in a fasting (5-h) and non-fasting regime, insulin response, and intraperitoneal glucose tolerance test (ipGTT). For the evaluation in a fed state, blood glucose was measured early in the morning (8 a.m.), and then intraperitoneal insulin administration was performed (0.75 IU/kg, Novorapid 100 IU/mL, Novo Nordisk A/S, Bagsvaerd, Denmark), with a blood glucose measurement after 15 min. The insulin response was assessed between the groups by subtracting the blood glucose values 15 min after insulin injection from the ones at baseline (t = 0) (delta value: Δ). For the ipGTT, mice underwent a 5-h morning fasting, and blood glucose levels were measured using a FreeStyle Lite Blood Glucose Meter (Abbott Diabetes Care Inc., Alameda, CA, USA) at 0, 15, 30, 60, and 120 min following an intraperitoneal injection of a 50% D-glucose solution (2 g/kg body weight).

### 2.3. Cardiovascular Function

*Blood Pressure*: The Coda High Throughput Noninvasive Tail Monitoring System (Kent Scientific, Torrington, CT, USA) was utilized for conscious blood pressure monitoring, following the company’s guidelines. This system employs Volume Pressure Recording (VPR) to gauge blood pressure by detecting tail blood volume. Systolic (SAP), diastolic (DAP), and mean arterial blood pressure (MAP) were assessed through 45 cycles, and data were averaged from each animal utilized for analysis.

*Heart Injury*: As an indicator of myocardial injury, cardiac troponin I (CtnI) levels were measured in plasma samples using a colorimetric sandwich ELISA assay (No. NBP3-00456, Novous Biologicals, CO, USA). As a marker of inflammation, interleukin 6 (IL-6) was quantified by the IL-6 Mouse ProQuantum Immunoassay Kit (Catalog #A43656—Invitrogen, Stockholm, Sweden).

*Vessel Reactivity*: After removing the adipose and connective tissues around the mesenteric arteries, the vessels were cut into segments of approximately 2 mm in length. These segments were then mounted on a wire myograph system (Model 620M; Danish Myo Technology, Hinnerup, Denmark) using 25 µm wire, which was connected to a Powerlab system (Powerlab 4/30) for recording isometric tension. The chambers were pre-filled with 8 mL of physiological salt solution (PSS) at 37 °C and pH 7.4, and aerated with carbogen gas (95% O_2_, 5% CO_2_). After mounting, a loading force of 2.5 mN was applied to simulate near-physiological pressure, and the vessels were allowed to equilibrate for 45 min. The vessel segments were then contracted using 75 mM of potassium chloride (KCl) combined with 10 μM of phenylephrine (PE) to assess the maximum contractile response of the vascular smooth muscle cells. Following three washes, the vessels were normalized for 40 min before performing concentration–response curves for PE (0.1 nM to 10 μM), acetylcholine (Ach, 0.1 nM to 100 μM), and sodium nitroprusside (SNP, 0.1 nM to 10 μM). An initial contraction, at around 70% of the maximum contraction, was induced by PE before the addition of Ach or SNP to examine endothelial-dependent or -independent vasorelaxation, respectively.

### 2.4. Renal Function

*Glomerular Filtration Rate (GFR)*: Renal function was assessed by analyzing the clearance kinetics of plasma FITC-conjugated inulin. A 1% FITC–inulin solution (Sigma-Aldrich F3272, Sigma-Aldrich, St. Louis, MO, USA) in phosphate-buffered saline (PBS) was prepared and sterilized using a 0.22-μm filter. Approximately 100 µL of this solution was injected into the tail vein of each animal, and blood samples were collected in heparinized capillaries at 1, 3, 5, 10, 15, 35, 55, and 75 min post-injection. The blood samples were then centrifuged to isolate plasma, which was buffered to pH 7.4 with 500 mM of HEPES in PBS. Fluorescence intensity was measured using a multi-mode microplate reader (SpectraMax iD3, Molecular Devices, San Jose, CA, USA) with an excitation wavelength of 480 nm and emission at 530 nm. A standard curve was generated by testing known concentrations of FITC–inulin, establishing a linear correlation between the fluorescence intensity and inulin concentration.

*Albuminuria*: Urinary albumin level, a biomarker for early-stage chronic kidney diseases, was analyzed using a fluorometric assay kit (No. ab241017, Abcam, Cambridge, UK). Urinary specific gravity was measured by refractometry, employing an analog refractometer (KERN ORA-P, Zollernalbkreis, Germany). All procedures in the protocol were performed according to the manufacturer’s instructions.

*Mitochondrial oxygen efficiency*: Mitochondria from the kidney were isolated by differential centrifugation, as described elsewhere [35]. Mitochondrial function was analyzed by using high-resolution respirometry (Oroboros, O2k, Innsbruck, Austria). Mitochondrial integrity was verified by measuring the respiratory control ratio (RCR), defined as the ratio of maximal complex I-mediated respiratory capacity (state 3) with pyruvate (5 mM), malate (2 mM), and ADP (2.5 mM) to the leak state respiration without adenylates (using pyruvate and malate). Mitochondrial oxygen efficiency (the P/O ratio) was determined in the presence of pyruvate (5 mM) and malate (2 mM) by a steady-state infusion of non-saturating levels of ADP using the TIP2K titration injection micropump. The quantity of mitochondria in the chamber was adjusted to reach a respiration rate during ADP infusion that corresponds to approximately 50% of the maximal complex I-dependent state 3 respiration. The P/O ratio was calculated by dividing the rate of ADP infusion by the oxygen consumed during steady-state respiration. Background correction was applied to account for the oxygen present in the ADP solution.

### 2.5. Histopathological Evaluation

Kidney and gonadal white adipose tissue (gWAT) were collected and fixed with 4% paraformaldehyde for histopathological analysis. After fixation, samples were embedded in paraffin and sectioned (5 µm). The kidney sections were stained with hematoxylin–eosin (HE) and periodic acid–Schiff (PAS). The adipose tissue was stained with only hematoxylin–eosin. The prepared sections were evaluated in a blinded fashion by a specialized histopathologist.

Tissue morphology, including fibrosis, necrosis, and inflammatory infiltration, was examined and quantified. A scoring system was used to assess tubular damage based on the distribution of previously noted alterations, such as tubular vacuolization, dilation, atrophy, loss of brush border, and thickening of the tubular basement membrane [36].

Glomerular injury was assessed based on characteristics such as glomerular dilation, glomerulosclerosis, mesangial hypercellularity, matrix expansion, and irregular thickening of the glomerular basement membrane, examined in 8 random fields per sample. Adipocyte size in gWAT was measured as the cross-sectional area of 300 adipocytes per mouse and fat depot using the Adiposoft plugin in Fiji ImageJ software (version 1.16, updated 8 April 2019). For each section, 50 cells were analyzed from six randomly selected areas, chosen blindly, followed by manual verification to ensure only intact adipocytes were included in the semi-automated analysis. Adipocyte size was limited to 30–120 µm for gWAT and 20–120 µm for scWAT to focus specifically on adipocytes. All images were captured using an Axioscope Microscope with an Axiocam 208 color camera (Carl Zeiss Microscopy, Stockholm, Sweden).

### 2.6. Statistics

Data were initially analyzed for normal distribution using the following tests: (1) the Anderson–Darling test; (2) D’Agostino–Pearson test; (3) Shapiro–Wilk test; and (4) Kolmogorov–Smirnov test, as recommended by the statistical software. For normally distributed data, results were described as mean ± SD, and analysis of variance (ANOVA) followed by Tukey’s multiple comparisons test was performed, as recommended by the statistical software used (GraphPad Prism version 9.2.0). Data that were not normally distributed were described as medians with interquartile ranges; for comparisons between these groups, the Kruskal–Wallis test was performed, followed by Dunn’s multiple comparisons test, as recommended by the statistical software used (GraphPad Prism version 9.2.0). Repeated, two-way ANOVAs, followed by Sidak post-hoc tests, were used to compare dose responses of isolated vessels (myograph). The half-maximal effective concentration (EC50) values for vessel reactivity were analyzed using non-linear regression. A *p* < 0.05 was considered statistically significant. In the figures, the following symbols denote statistical significances: * for *p* < 0.05, ** for *p* < 0.01, *** for *p* < 0.001, and **** for *p* < 0.0001. Non-significant differences are indicated as ns.

## 3. Results

### 3.1. Flexovital Improves Metabolic Parameters in Mice with UNX+WD

Despite being exposed to a diet high in fat and sugar, the animals did not exhibit the typical significant weight gain commonly associated with a standard Western diet (WD) (Figure 2A). In this experiment, a modified WD enriched with salt was used, and previous studies have reported that salt can limit weight gain, consistent with the findings observed here (Figure 2B). Moreover, the impact of salt on fat metabolism is currently under intense investigation, given its critical role in the pathophysiology of various metabolic diseases [37,38].

Despite similar body weights across all experimental groups, weight gain was 22% less pronounced in the UNX+WD+FLX group compared to the UNX+WD group (*p* < 0.05) (Figure 2B). Animals subjected to this model exhibited significant alterations in body composition, which were mitigated by FLX supplementation. These changes included a reduction in fat mass (Figure 2C) and stable lean mass (Figure 2D), resulting in significant differences in the fat-to-lean mass ratio (Figure 2E). These findings suggest a potential long-term benefit of FLX supplementation. Histopathological analysis of adipose tissue supported the in vivo body composition data (Figure 2B–E), with a non-significant reduction in adipocyte size following FLX treatment compared to the UNX+WD group (Figure 2F and Figure 3).

As expected, animals in the sham group fed a normal diet exhibited lower values on the glucose tolerance test (ipGTT). Among the groups on a WD, no significant differences were observed (Figure 4A). However, FLX supplementation showed a positive effect, reflected in a reduced area under the ipGTT curve (Figure 4B), lower fasting blood glucose levels (Figure 4C), and improved response to insulin administration (Figure 4D).

### 3.2. Flexovital Improves Cardiovascular Health in Mice with UNX+WD

Exposure to a high-salt WD in animals with a 50% reduction in nephron number early in life (UNX) led to a significant increase in blood pressure (Figure 5A–C). However, FLX supplementation resulted in a significant reduction in mean arterial pressure, as well as systolic and diastolic blood pressure (Figure 5A–C), without affecting heart rate (Figure 5D). Additionally, the WD-UNX group showed a 2–3-fold increase in serum concentrations of the cardiac injury biomarker troponin I (Figure 5E), indicating cardiac cytotoxicity and troponin leakage. In contrast, this elevation in troponin was not as obvious in the FLX-supplemented group, whose levels tended to be lower than in the UNX+WD group (*p* = 0.10) and similar to the sham group (Figure 5E). Interestingly, the FLX group exhibited numerically lower IL-6 levels compared to the UNX+WD group (0.03 ± 0.05 vs. 0.14 ± 0.26). However, this difference was not statistically significant, and further investigation is needed to explore this observation in greater detail.

FLX supplementation was also linked to better endothelial function in the UNX+WD group. While no differences were observed among the groups in their contractile responses to phenylephrine (Figure 6A,B), mice in the UNX+WD group exhibited significant endothelial dysfunction. This impairment was markedly improved by FLX treatment, as demonstrated by enhanced endothelium-dependent responses to acetylcholine in the myograph system (Figure 6C). Additionally, no significant differences were found among the groups in their EC50 values for acetylcholine (Figure 6D).

### 3.3. Flexovital Improves Kidney Function and Reduces Glomerular and Tubular Injuries in Mice with UNX+WD

The cardiovascular-kidney-metabolic syndrome model was associated with renal dysfunction and increased glomerular injury after 12 weeks, as previously described by Carvalho et al. [34]. Chronic supplementation with FLX, initiated at the onset of the experimental CKM model, was associated with improved renal mitochondrial efficiency (Figure 7A), as reflected by the higher P/O ratio, and preservation of glomerular filtration rate in the FLX group (Figure 7B). Additionally, FLX improved various histopathological and morphological markers of renal health, including reduced glomerular injury without affecting glomerular area (Figure 7C,D and Figure 8), attenuation of tubular injury (Figure 7E and Figure 8), and a clear trend towards reduced albuminuria (Figure 7F).

## 4. Discussion

In this study, we investigated the therapeutic potential of the food additive Flexovital (FLX) in a recently developed murine cardiovascular-kidney-metabolic (CKM) disease model. This model, featuring a 50% reduction in renal mass (via UNX), combined with a Western diet (WD) high in fat, sugar, and salt, led to notable cellular, organ, and functional damage [34]. CKM syndrome mice (UNX+WD) displayed increased body fat, endothelial dysfunction, hypertension, cardiac injury with troponin leakage, renal dysfunction and injuries, and mitochondrial impairment, as well as albuminuria, compared to sham-operated mice on a standard diet. Supplementation with a specially formulated nitric oxide-enhancing food additive (i.e., FLX)—containing *Rhodiola rosea* and beetroot extracts, and selected amino acids—partially or fully prevented many of these CKM-related pathologies.

Mice treated with FLX showed a significant reduction in body weight gain (22%) compared to the untreated UNX+WD group. The fat-to-lean mass ratio increased by almost 20% (*p* < 0.01) and the adipocyte area by more than 30% (*p* < 0.001) in the UNX+WD group, but both were nearly restored to normal levels in the FLX-treated group (*p* < 0.01). Additionally, FLX treatment in the UNX+WD group showed a trend toward reduced systemic inflammation, as indicated by IL-6 measurements, though the results were not conclusive due to a lack of sufficient samples for analysis.

In the UNX+WD group, a significant increase in blood pressure (MAP, SAP, and DAP) was observed, along with reduced endothelium-dependent vasorelaxation. These effects may be partially attributed to a marked reduction in glomerular filtration rate (GFR) and elevated levels of dimethylarginines (ADMA and SDMA), which have been previously reported [28,29]. Additionally, cardiac injury, as indicated by plasma troponin I levels, showed a nominal 2–3 fold increase in the UNX+WD group but appeared to be almost normalized with FLX treatment. The contribution of the specially formulated diet to the development of hypertension and endothelial dysfunction remains uncertain; however, components such as high salt content are likely contributing factors.

FLX pretreatment significantly mitigated both the rise in blood pressure and the impairment of endothelium-dependent vasodilation in this model. This protective effect may be linked to the inclusion of substrates that facilitate nitric oxide bioactivity, such as L-arginine, L-citrulline, and beetroot extract, as well as a potential inhibitory action on phosphodiesterase type 5 (PDE5) by *Rhodiola*, as discussed further below. Overall, our findings demonstrate a preventive effect of FLX treatment on cardiovascular, kidney, and metabolic functions in this CKM syndrome model, which is characterized by moderate renal dysfunction and a Western diet.

GFR was markedly decreased in the UNX+WD group but was significantly improved by chronic FLX supplementation (*p* < 0.05). The UNX+WD group exhibited significant glomerular injuries with mesangial proliferation (*p* < 0.001), which were less pronounced in the FLX group (*p* < 0.001). The tubular injury score (ranging from 1 to 10) increased from 1 in the sham group to 6.6 in the UNX+WD group and was partially ameliorated to 4.5 in the FLX group (*p* < 0.05). Renal function, compromised by the WD, resulting in glomerular and tubular damage, was substantially preserved in the FLX-treated group.

In the disease model in this study, the combination of partial renal function loss with a WD accelerated cardiovascular, kidney, and metabolic dysfunction. This study does not specify the individual contributions of UNX and WD, though both clearly exacerbate these conditions. Sham-operated mice on the same diet experienced less vascular and renal damage over three months [34]. While UNX alone typically resulted in only a 35% reduction in GFR due to hemodynamic changes and compensatory hypertrophy of the contralateral kidney [39,40]. WD+UNX mice showed a 60% reduction in GFR, with significant glomerular and tubular damage. These abnormalities, rarely seen with UNX alone over the same period, suggest the WD plays a key role that warrants further investigation to distinguish the effects of UNX and WD.

Previous studies using UNX, high salt intake, or WD models of hypertension, kidney disease, or metabolic syndrome suggest the beneficial effects of dietary L-arginine [28,41], L-citrulline [42], beetroot [29,43,44], or *Rhodiola rosea* [45], which are all key components of the FLX supplement. While the exact mechanisms behind FLX’s protective effects are not yet clear, they are likely multifactorial. Considering its composition (favoring nitric oxide bioavailability), a more stabilized vasorelaxation ability likely plays a significant role. This effect may be linked to the presence of L-arginine and L-citrulline in FLX, which serve as substrates for nitric oxide production. Other studies have suggested that *Rhodiola rosea* via yet unknown mechanisms could prolong nitric oxide signaling by reducing the degradation of downstream cGMP, thereby enhancing vasodilation beyond the mere addition of substrates for nitric oxide generation [46]. Improved endothelial-dependent vasodilation may benefit microcirculation, which is often compromised in renal failure and likely exacerbated by the unhealthy diet in the UNX+WD group. This exacerbation is associated with increased blood pressure, pro-inflammatory changes, and oxidative stress. In this study, FLX clearly seemed to enhance endothelial function and, subsequently, reduce blood pressure, as evidenced by the significant improvements observed in the treated group.

The organ-protective effects of FLX may result from a healthier microcirculatory environment, as discussed above. Additionally, bioactive components in *Rhodiola rosea* (e.g., salidroside) and beetroot (e.g., betanin and inorganic nitrate) may upregulate nuclear factor erythroid 2-related factor 2 (Nrf2) expression or activation, potentially exerting both anti-oxidative, anti-inflammatory, and cytoprotective effects [46,47,48,49,50], though Nrf2 expression was not measured in this study. The specific components of the *Rhodiola rosea* and beetroot extracts contributing to the observed cellular- and organ-protective effects, as well as improvements in glucose and lipid metabolism, remain to be elucidated. Additionally, the favorable effects of FLX on body weight gain, visceral adiposity, and glucose metabolism may indicate an influence on GLP-1 receptor function similar to pharmaceutical treatments for diabetes and obesity. While these mechanisms were unexplored in the present study, it will be a central focus of future research.

***Strengths and limitations***: A limitation of the study is that the use of a combined formula makes it difficult to pinpoint the mechanism of action; in particular, the contributions of the individual components present in the formulation. Moreover, in this experimental study, FLX was administered at a dose relevant to humans, and showed preventive effects on CKM disturbances. However, its ability to reduce or reverse already established injuries remains unclear. In the same model, CKM dysfunctions progressed over 20 weeks, and future studies are planned to investigate whether FLX can mitigate or even reverse these pathologies compared to placebo. These studies will also aim to further explore the underlying mechanisms of action and examine the individual contributions of diet and UNX to disease development. Finally, the current study was conducted in mice, and the translation of these findings to humans is still uncertain; future investigations are being planned to assess the preventive effects of FLX in human subjects.

## 5. Conclusions and Future Perspectives

In this study, using a cardiovascular-kidney-metabolic (CKM) syndrome model, mice with reduced nephron numbers, due to a unilateral nephrectomy, developed significant metabolic, cardiovascular, and renal dysfunction when chronically fed a Western diet high in fat, carbohydrates, and salt. Notably, Flexovital supplementation demonstrated substantial protective effects across most functional parameters evaluated in the UNX+WD-induced CKM syndrome model. Future clinical trials are needed to confirm the promising preventive and therapeutic effects of Flexovital in humans with cardiovascular-kidney-metabolic syndrome.

## Figures and Tables

**Figure 1 nutrients-16-04105-f001:**
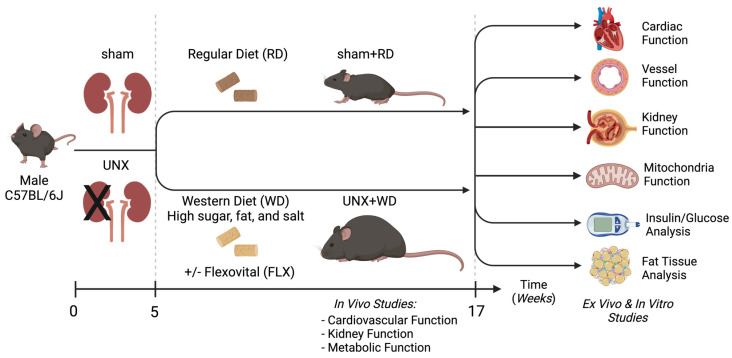
Illustration of the experimental study design. At the age of 4 weeks, mice underwent unilateral nephrectomy (UNX) or sham surgeries. Following one week of recovery, mice were chronically fed with a regular rodent diet (RD) or a special Western diet (WD), which contained not only high fat and sugar but also high salt. For the treated groups, the Flexovital supplement was added to the water at a dose of 0.25 mg/mL. In vivo assessments of cardiovascular, kidney, and metabolic functions were performed, and thereafter, the animals were euthanized, and tissue samples were collected and prepared for various analyses, including both functional and biochemical studies, as well as histological analysis. Illustration created in BioRender. Carlstrom, M. (2024) BioRender.com/k37u826 (accessed on 20 November 2024).

**Figure 2 nutrients-16-04105-f002:**
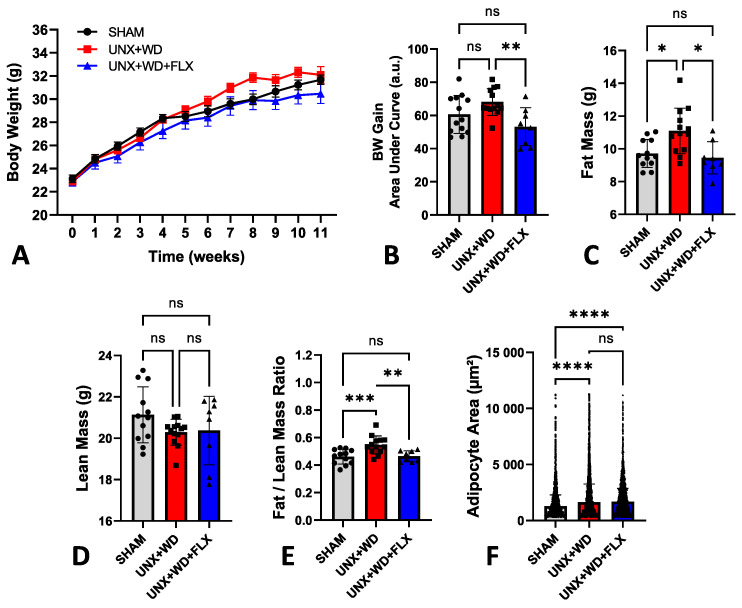
Analysis of body composition and body weight of animals submitted to sham surgery or to unilateral nephrectomy (UNX) and chronic feeding with Western diet (WD) with or without Flexovital (FLX) supplementation. (**A**) Body weight; (**B**) body weight gain; (**C**) fat mass in grams; (**D**) lean mass in grams; (**E**) ratio between fat and lean mass; (**F**) adipocyte area evaluated by specific histopathological software. Data in panels (**A**–**E**) were compared by an ANOVA, followed by Tukey’s mean comparison test. Data in panel (**F**) was compared by a Kruskal–Wallis test followed by Dunn’s multiple comparisons test. * for *p* < 0.05, ** for *p* < 0.01, *** for *p* < 0.001, and **** for *p* < 0.0001. Non-significant differences are indicated as ns.

**Figure 3 nutrients-16-04105-f003:**
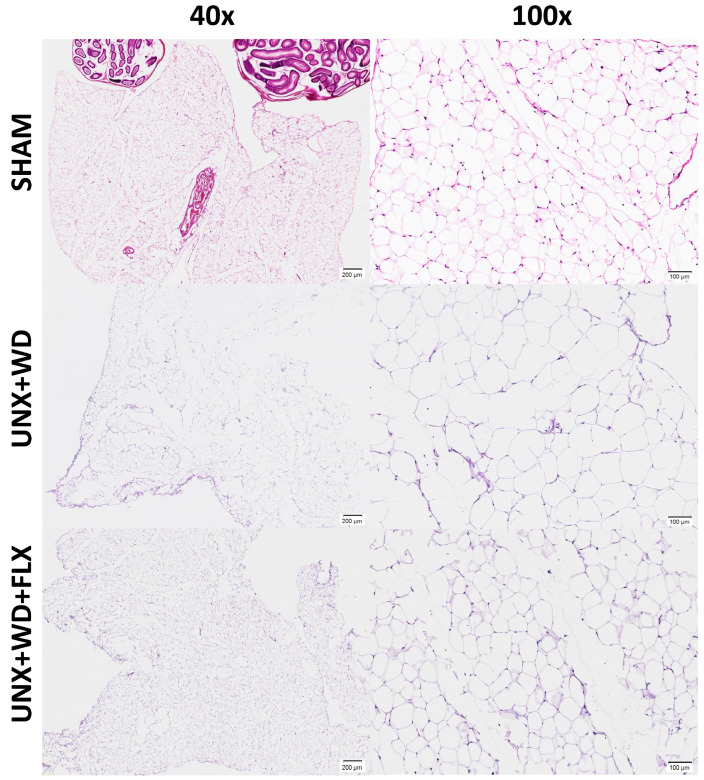
Effects of Flexovital (FLX) on adipocytes. Consistent with the in vivo body composition analysis, examination of adipose tissue from white fat stores ex vivo revealed an enlarged adipocyte area in mice with unilateral nephrectomy (UNX) and consumption of a Western diet (WD) (second line) compared to controls (first line). However, even with a reduction in the average size of adipocytes, statistical analysis revealed no differences between the group supplemented with Flexovital and the disease model (third line). Samples stained with hematoxylin–eosin: first column 40× magnification, second column 100× magnification.

**Figure 4 nutrients-16-04105-f004:**
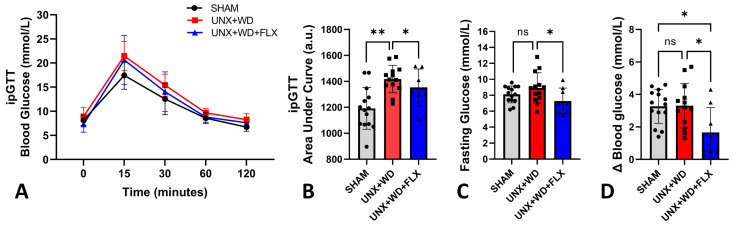
Glucose metabolism analysis of animals submitted to sham surgery or to unilateral nephrectomy (UNX) and chronic feeding with Western diet (WD) with or without Flexovital (FLX) supplementation. (**A**) Intraperitoneal glucose tolerance test (ipGTT) curve; (**B**) area under the ipGTT curve; (**C**) 5-h fasting blood glucose (morning fasting); (**D**) delta value of blood glucose subtraction before and after insulin injection. The data in all panels were compared by ANOVA followed by Tukey’s mean comparison test. * for *p* < 0.05, ** for *p* < 0.01. Non-significant differences are indicated as ns.

**Figure 5 nutrients-16-04105-f005:**
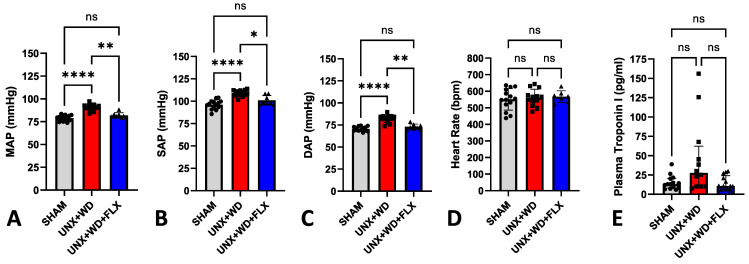
Cardiovascular effects of Flexovital (FLX) in animals submitted to sham surgery or to unilateral nephrectomy (UNX) and chronic feeding with Western diet (WD) with or without Flexovital (FLX) supplementation. (**A**) Mean arterial pressure (MAP); (**B**) systolic blood pressure (SAP); (**C**) diastolic blood pressure (DAP); (**D**) heart rate; (**E**) plasmatic concentrations of troponin I. The data in panels (**A**–**D**) were compared by ANOVAs followed by Tukey’s mean comparison test. The data in panel (**E**) were compared by the Kruskal–Wallis test followed by Dunn’s multiple comparisons test. * for *p* < 0.05, ** for *p* < 0.01, and **** for *p* < 0.0001. Non-significant differences are indicated as ns.

**Figure 6 nutrients-16-04105-f006:**
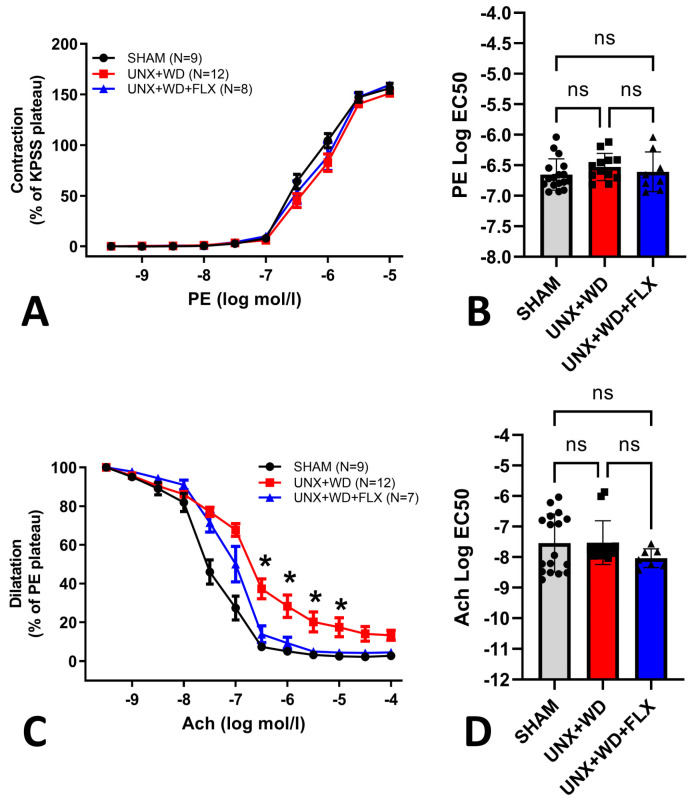
Vascular reactivity effects of Flexovital (FLX) in animals submitted to a sham surgery or to a unilateral nephrectomy (UNX) and chronic feeding with Western diet (WD) with or without Flexovital (FLX) supplementation. (**A**,**B**) Dose–response curve of phenylephrine (PE)-induced vascular contraction, followed by EC50 comparisons; (**C**,**D**) dose–response curve of acetylcholine (Ach)-induced vascular relaxation, followed by EC50 comparisons. The data in panels (**A**,**C**) were compared by repeated two-way ANOVAs, followed by a Sidak post-hoc test. The data in panels (**B**,**D**) were analyzed using non-linear regressions and compared with a Kruskal–Wallis test followed by Dunn’s multiple comparisons test. * for *p* < 0.05. Non-significant differences are indicated as ns.

**Figure 7 nutrients-16-04105-f007:**
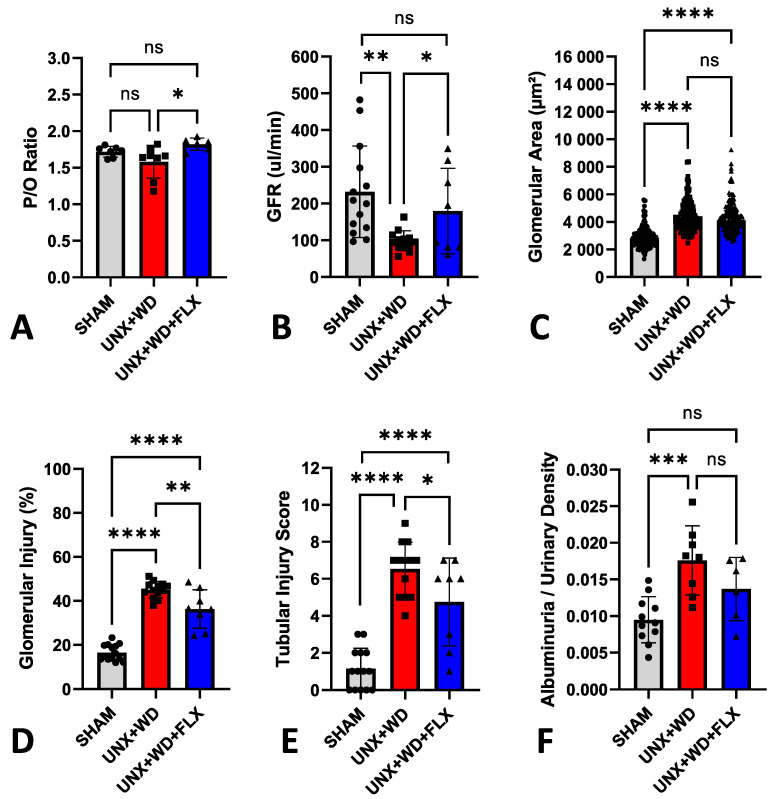
Renal effects of Flexovital (FLX) in animals submitted to a sham surgery or to a unilateral nephrectomy (UNX) and chronic feeding with a Western diet (WD) with or without Flexovital (FLX) supplementation. (**A**) Phosphate/oxygen ratio (P/O ratio), calculated by oxygraph, refers to the amount of ATP produced from the movement of two electrons through a defined electron transport chain terminated by the reduction of an oxygen atom; (**B**) glomerular filtration rate (GFR); (**C**) glomerular area; (**D**) percent glomerular injury; (**E**) renal tubular injury score; (**F**) urinary albumin corrected for urine-specific gravity. The data in panel (**C**) were compared by a Kruskal–Wallis test followed by Dunn’s multiple comparisons test, and the data in all other panels were compared by ANOVAs, followed by Tukey’s mean comparison test. * for *p* < 0.05, ** for *p* < 0.01, *** for *p* < 0.001, and **** for *p* < 0.0001. Non-significant differences are indicated as ns.

**Figure 8 nutrients-16-04105-f008:**
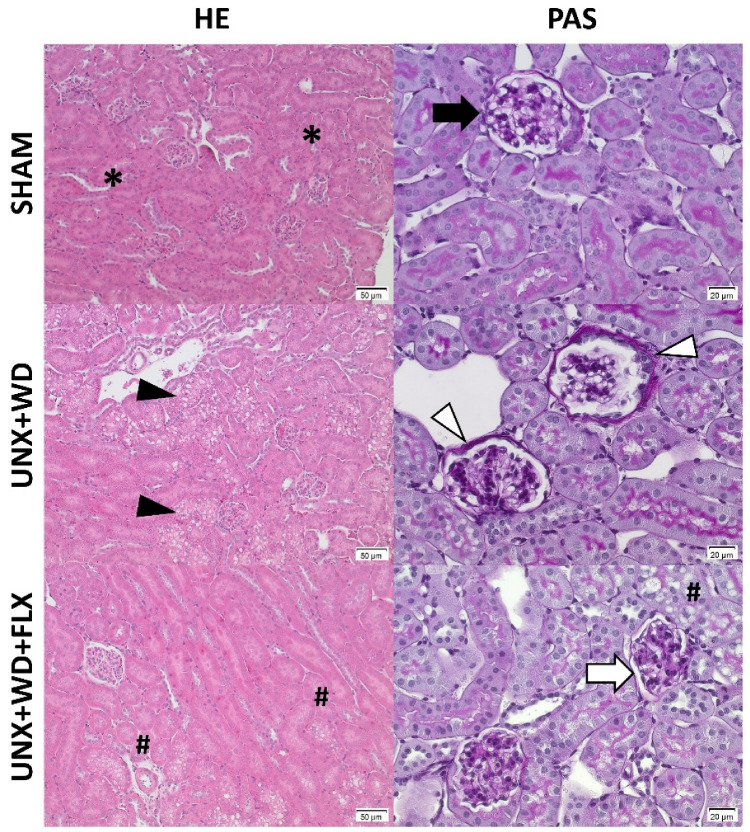
Histopathological evaluation of kidney samples from animals submitted to a sham surgery or to a unilateral nephrectomy (UNX) and chronic feeding with a Western diet (WD) with or without Flexovital (FLX) supplementation. First column: samples stained with hematoxylin–eosin and photographed at 200× magnification. Second column: samples stained with periodic acid–Schiff (PAS) and photographed at 200× magnification. First line: samples from the sham group with preserved integrity of the cortical tubules (*), intact glomeruli, with a normal diameter and appearance, showing preservation of the membrane (black arrow) and the absence of fibrosis and inflammatory infiltrates. Second line: samples from the UNX+WD group with diffuse vacuolar alterations in the cortical tubules (black arrowhead) and increased glomerular membrane thickness without glomerular and tubular dilatation (white arrowhead). Third line: samples from the FLX group with considerable reduction in vacuolar alterations in the tubules (#) and reduction in the thickness of the glomerular membrane (white arrow), suggesting beneficial morphological alterations associated with FLX and a general appearance similar to that observed in the sham group.

## Data Availability

The datasets used and/or analyzed during the current study are available from the corresponding author upon reasonable request.

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
