# Peer review of "Protective Effects of the Food Supplement Flexovital in a Model of Cardiovascular-Kidney-Metabolic Syndrome in Mice"

_nutrients, 2024, doi:10.3390/nu16234105_

Round 1

Reviewer 1 Report

Comments and Suggestions for Authors

This is an interesting research article with adequate novelty. Some poind should be addressed.

- In the Introdiction section, the authors should provide more information about Flexovital supplemment.

- The authors should clearly report the whole number of the experimental animals of our study.

- Figure 2 is not easily readable. It could be split into two figures.

- The resolution and the quality performance - contrast of Figure 3 should be improved.

- Figure 4 is not easily readable. It could be split into two figures.

Author Response

Reviewer 1

AUTHORS’ COMMENT: We sincerely thank the reviewers for their thoughtful comments and constructive feedback, which have significantly contributed to improving the quality of our manuscript. Below, we have responded point-by-point to all the comments/suggestions.

This is an interesting research article with adequate novelty. Some points should be addressed.

- In the Introdiction section, the authors should provide more information about Flexovital supplemment.

AUTHORS’ RESPONSE: Thank you for this comment. In the revised Introduction (last paragraph) we have included more details about the food supplement Flexovital and its different components and the doses used in humans:

Flexovital, administered as capsules (up to a maximum of 4 per day in humans), contains Rhodiola rosea extract (100 mg), beetroot extract (100 mg), and the amino acids L-arginine (175 mg) and L-citrulline (125 mg), along with small amounts of magnesium (25 mg) and vitamin C (25 mg) per capsule.

- The authors should clearly report the whole number of the experimental animals of our study.

AUTHORS’ RESPONSE: In the revised manuscript (Section 2.1. Animals and Experimental Design) we have reported the total number of experimental animals used in this study (n=35).

- Figure 2 is not easily readable. It could be split into two figures.

AUTHORS’ RESPONSE: Thank you for this suggestion. In the revised manuscript, we have divided this figure into two parts. The new Figure 2A-F presents the changes in body composition alongside the adipocyte area. The histology photomicrographs of the adipose tissue have been moved to a new figure (Figure 3). Additionally, the new Figure 4A-D illustrates the data on glucose metabolism and homeostasis. We hope this updated presentation improves the clarity and readability of the data.

- The resolution and the quality performance - contrast of Figure 3 should be improved.

AUTHORS’ RESPONSE: Thank you for the comment. In the updated version of the manuscript, we have provided the figure with improved resolution and made minor adjustments to brightness and contrast, ensuring that the original raw data remains unaffected.

- Figure 4 is not easily readable. It could be split into two figures.

AUTHORS’ RESPONSE: Thank you for this suggestion. In the revised manuscript, we have divided this figure into two parts. The new Figure 5 A-E presents the changes in blood pressure and heart rate alongside the plasma troponin levels. Additionally, the new Figure 6 A-D illustrates the data on vascular reactivity. We hope this updated presentation improves the clarity and readability of the data.

Reviewer 2 Report

Comments and Suggestions for Authors

The experiments were carefully designed and performed, but the statistical analysis is deplorable. For undisclosed reasons, the authors assumed that their data follow a normal distribution and therefore used ANOVA, Tukey’s multiple comparisons test, and t test for statistical analysis and mean ± SD for data characterization. They should first check data distribution and then choose the appropriate statistical test and the adequate manner of characterizing data. Until then the paper is statistically invalid.

More than 20 sets of data derived from these experiments have undergone statistical analysis. We are told that a P<0.05 (i.e. 1 in 20) was considered statistically significant - this is statistically incorrect. Given the 0.05 threshold for statistical significance it is very likely that at least one (20 divided by 20) of the results has a less than 0.05 p-value by pure chance. A Bonferroni correction should be applied and therefore a 0.05 / 20 = 0.0025 threshold for statistical significance should be employed. Otherwise, whatever experiment out of which at least 20 sets of data emerge will yield at least one statistically significant result by sheer luck - this is p-hacking, not scientific research.

Author Response

Reviewer 2

The experiments were carefully designed and performed, but the statistical analysis is deplorable. For undisclosed reasons, the authors assumed that their data follow a normal distribution and therefore used ANOVA, Tukey’s multiple comparisons test, and t test for statistical analysis and mean ± SD for data characterization. They should first check data distribution and then choose the appropriate statistical test and the adequate manner of characterizing data. Until then the paper is statistically invalid.

More than 20 sets of data derived from these experiments have undergone statistical analysis. We are told that a P<0.05 (i.e. 1 in 20) was considered statistically significant - this is statistically incorrect. Given the 0.05 threshold for statistical significance it is very likely that at least one (20 divided by 20) of the results has a less than 0.05 p-value by pure chance. A Bonferroni correction should be applied and therefore a 0.05 / 20 = 0.0025 threshold for statistical significance should be employed. Otherwise, whatever experiment out of which at least 20 sets of data emerge will yield at least one statistically significant result by sheer luck - this is p-hacking, not scientific research.

AUTHORS’ RESPONSE: We thank the reviewer for the detailed feedback regarding our statistical methods. We appreciate the constructive critique and have made revisions accordingly to enhance the rigor of our analysis.

Assumptions of Normal Distribution: We acknowledge the reviewer’s concerns about our assumption of normality in the data. In response, we have conducted additional tests to assess the distribution of our datasets, including the Shapiro-Wilk test (or other appropriate tests). Most parameters tested were normally distributed. However, the following parameters did not meet the assumptions of normality:

New Figure and Panel

Parameter

Figure 2 F

Adipocyte area

Figure 5 E

Plasma Troponin

Figure 6 B

PE contraction (EC50)

Figure 6 D

Ach dilatation (EC50)

Figure 7 C

Glomerular area

For these datasets, we reanalyzed the results using a non-parametric test (i.e., the Kruskal-Wallis test). Of these reanalyzed parameters, only the comparisons in Figure 2F and Figure 5E failed to reach statistical significance, though they demonstrated a trend. These revisions have been reflected in the Methods (Section 2.6: Statistics) and Results sections of the manuscript.

Data Characterization: We also appreciate the reviewer’s recommendation for a more precise approach to data characterization. Accordingly:

  • For non-normally distributed parameters (see Table above), we have updated the manuscript to present the data as median with interquartile range (IQR).
  • For normally distributed parameters, we have retained the presentation as mean ± SD, as this remains statistically appropriate.

The manuscript figures and legends have been revised to incorporate these changes, ensuring greater statistical accuracy and clarity.

Multiple Comparisons and Correction of p-values: We thank the reviewer for raising concerns about the potential risk of Type I errors given the use of multiple comparisons. To address this, we employed GraphPad Prism (version 9.2.0) for all statistical analyses, including post hoc tests. For example, when conducting ANOVA followed by Tukey's multiple comparisons test, the software automatically adjusts for multiple comparisons. Tukey’s test controls the familywise error rate (FWER), ensuring that the overall probability of a Type I error across all comparisons remains at the designated significance level (e.g., α = 0.05).

We believe that parameter-specific multiple comparison corrections, as performed in our study, are preferable when analyzing independent outcomes. This approach strikes a better balance between Type I and Type II errors compared to a study-wide correction, which could be overly stringent unless the parameters are tightly related or the analyses are exploratory within a single outcome.

To illustrate, a power analysis based on our study design (three experimental groups) and assumed mean values, standard deviations, and effect sizes indicates that achieving a desired significance level (e.g., p < 0.00067, as would be required by a study-wide correction) would necessitate an unfeasible group size of 200–300 mice. Such a requirement contradicts the 3R principle (Replacement, Reduction, Refinement) for ethical experimental animal studies and is not standard practice in the field.

In summary, we appreciate the reviewer’s feedback, which has helped us improve the statistical rigor of our work. We believe that the revisions described above address the reviewer’s major concerns and substantially strengthen the statistical validity of our study.
